# Limitations and Recommendations for Advancing the Occupational Therapy Workforce Research Worldwide: Scoping Review and Content Analysis of the Literature

**DOI:** 10.3390/ijerph19127327

**Published:** 2022-06-15

**Authors:** Tiago S. Jesus, Karthik Mani, Ritchard Ledgerd, Sureshkumar Kamalakannan, Sutanuka Bhattacharjya, Claudia von Zweck

**Affiliations:** 1Center for Education in Health Sciences, Institute for Public Health and Medicine, Feinberg School of Medicine, Northwestern University, Chicago, IL 60611, USA; 2Department of Occupational Therapy, School of Health Professions, The University of Texas Medical Branch at Galveston, Galveston, TX 77555, USA; kamani@utmb.edu; 3World Federation of Occupational Therapists (WFOT), 1211 Geneva, Switzerland; executivedirector@wfot.org (R.L.); opservices@wfot.org (C.v.Z.); 4Department of Social Work, Education and Community Wellbeing, Northumbria University, Newcastle upon Tyne NE7 7XA, UK; sureshkumar.kamalakannan@northumbria.ac.uk; 5Department of Occupational Therapy, Byrdine F. Lewis College of Nursing and Health Professions, Georgia State University, Atlanta, GA 30302, USA; sbhattacharjya@gsu.edu

**Keywords:** health workforce, health personnel, human resources for health, occupational therapists, rehabilitation, review

## Abstract

Occupational therapy workforce research can help determine whether occupational therapists exist in sufficient supply, are equitably distributed, and meet competency standards. Advancing the value of occupational therapy workforce research requires an understanding of the limitations and recommendations identified by these investigations. This scoping review and content analysis synthesizes the study limitations and recommendations reported by the occupational therapy research worldwide. Two independent reviews included 57 papers from the past 25 years. Stated limitations included: focus on cross-sectional studies with small and convenience samples; participants from single settings or regions; local markets or preferences not specified; focus on self-reported data and intentions (rather than behaviors or occurrences); challenges in aggregating or synthesizing findings from descriptive data; lack of statistical adjustment for testing multiple associations; and the lack of detailed, up-to-date, and accessible workforce data for continuous monitoring and secondary research. Stated recommendations included: strengthening routine workforce data collection; developing longitudinal studies that include interventions (e.g., recruitment or retention packages); developing context-sensitive comparisons; studying the impact on ultimate outcomes; promoting nation-wide, coordinated workforce plans and requirements; and fostering international coalitions for workforce research and developments at scale. These study limitations and recommendations reported by the literature must be considered in the design of a local and global occupational therapy workforce research agenda.

## 1. Introduction

The health workforce is one of the building blocks of health systems; workforce strength is essential for an effective and equitable coverage of population health needs [1,2]. Health workforce research can provide the evidence base to inform and evaluate population-centered workforce policies as well as workforce development practices [3,4,5]. Health workforce research can identify supply shortages, either current or forecasted, investigate inequitable human resources distributions, and study the impact of workforce policies, management, and regulations on human resources recruitment, retention, resilience, and performance [1,3,4,6].

Occupational therapists are health professionals that meet health, rehabilitation, and occupational needs of individuals experiencing a range of health conditions and disabilities [7,8]. To fulfill their professional role for health and wellbeing, occupational therapists need to be in sufficient supply, equitably distributed, motivated, and meet key competency standards [7]. However, the occupational therapy workforce worldwide has not been investigated regarding its scope and limitations.

Toward informing on a global strategy for the occupational therapy workforce research, the World Federation of Occupational Therapists (WFOT) initiated a multi-pronged scoping review to map the occupational therapy workforce research worldwide [9]. Findings published in the first paper focused on quantitative trends included a minimal yearly growth in publications (i.e., 14 years for an additional yearly publication), a small fraction of papers focused on low- and middle-income countries (LMICs), a majority of cross-sectional and exploratory studies with standardized instruments or inferential statistics often unused, and a minority of papers (25–30%) with funding support and more advanced study methods [10]. In the second paper, the type of findings generated by the occupational therapy workforce research worldwide were synthesized, and no substantiative trends emerged apart from a focus on attractiveness and retention in Australia and on supply and demand in the US [11]. Overall, research programs were nearly absent and other contemporary health workforce research topics or approaches were under-addressed (e.g., racial/ethnic representation) or not addressed at all (e.g., task sharing and situational analyses) [11]. Here, in the last paper from this project, we aim to synthesize the limitations and future recommendations that were reported in the occupational therapy workforce research. The study addressed questions regarding what type of study and data limitations and recommendations are reported by the occupational therapy workforce research. Altogether, these results will help inform a consultative process on global directions for occupational therapy workforce research.

## 2. Materials and Methods

We developed a scoping review, which tackles exploratory research questions on broad or complex topics toward identifying key concepts, research methods, type of evidence, and/or gaps in a research field [12,13,14,15]. We followed the Arksey & O’Malley’s framework [12,13,16] and the Joanna Briggs Institute’s guidelines for conducting scoping reviews [17]. The scoping review protocol has been peer-reviewed and published [9].

### 2.1. Searches

Medline/PubMed, Web of Science, Scopus, CINAHL, PDQ-Evidence, and OTseeker were scientific databases systematically searched; a full search strategy for PubMed database was detailed in the study protocol [9] and guided the searches in the other databases. The search strategy was appraised against the Peer Review of Electronic Search Strategies guidelines [18] and was conducted in June 2021. The Appendix A provides the detailed search strategies for each database.

Official, research-based reports were also searched through keyword searches and identified through international institutional websites: World Health Organization; Health Workforce Research (European Public Health Association); WFOT; and regional organizations of occupational therapist associations. Finally, snowballing (e.g., reference lists of included papers) and key informants (i.e., representatives of WFOT member organizations) were used to find any additional references, after being supplied with a preliminary list of inclusions. Although the database searches were run in June 2021, the iterative snowballing searches and the key-informant recommendations enabled the inclusion of papers published in the first quarter of 2022.

### 2.2. Eligibility

Inclusion criteria included occupational therapy workforce research fitting at least one category of workforce research defined in the study protocol (see Table 1) [9]. These inclusion categories were informed by a WFOT position statement [7], a review of the rehabilitation workforce literature [19], the global strategy on the health workforce [2], and a reader of research on the human resources for health [3].

Exclusion criteria refer to studies on the education of occupational therapists from a curriculum or pedagogical perspective and occupational health studies. Methodologically, we included quantitative, qualitative, or mixed-methods research, case studies, and systematic reviews published in peer-reviewed journals or in official institutional venues. Papers on the occupational therapy workforce or with occupational therapists as participants were included, even if other workers were included, but the papers needed to provide comparative or stratified results for occupational therapists. No language restrictions were applied, [9] yet only papers reported in English (or cumulatively in English and other language for journals publishing articles in more than one language) were identified. Exclusions were editorials, commentaries, letters, posters, study protocols, databases, or papers without a study question, replicable methods, or interpretation from the results.

Two independent reviewers (T.S.J. and K.M.) conducted titles and abstracts screening and full-text reviews, after an 80% or greater agreement in pilot tests on at least 5% of the references. Up to two discussion rounds among the reviewers were used for consensus on the eligibility decisions. No limits on geographic areas or timing of publication were used; yet a posteriori, as planned in the protocol [9], we applied a temporal cut-off determined by the research team at a saturation level. Hence, we only analyzed papers published in the last 25 years.

### 2.3. Data Extraction

We extracted the methodological features (e.g., study design, participants), geographic areas, settings, and key findings. A custom-built data extraction table was used for this process, including for extracting text quotations on (1) the stated limitations of the data or of the studies, and (2) the stated recommendations for workforce policies, practices, or future research. After a pilot test with 10% of the included references, one experienced reviewer (T.S.J.) extracted the information, fully verified by another research author (either K.M., S.K., or S.B.). Any disagreements among reviewers, especially involving adding other reported recommendations or limitations, were resolved through consensus, with no need to engage a third reviewer. Quality appraisals were not performed as common in scoping reviews [20,21].

### 2.4. Data Synthesis

We synthesized the stated limitations and reported recommendations from the literature based on a conventional content analysis [22], here inductive in nature, i.e., with categories emerging from the findings. The leading author (T.S.J.) performed the draft synthesis, iteratively edited by all other research authors. The order of categories, for either the stated limitations or reported recommendations, reflects the frequency of citation.

## 3. Results

Figure 1 provides the Preferred Reporting Items for Systematic Reviews and Meta-Analyses (PRISMA) flowchart of this review. From 1226 unique references identified, 57 papers were included after the temporal cut-off, i.e., published in the last 25 years. We synthesized below the types of (1) study and data limitations, and of (2) research recommendations.

### 3.1. Study and Data Limitations

#### 3.1.1. Cross-Sectional Studies with Convenience and Small Samples

This category was addressed by 19 papers. Convenience and small samples, in addition to low response rates, were frequently reported as a limitation, and affect the representativeness of the study population and, thereby, the validity and generalizability of the findings [23,24,25,26,27,28,29,30,31,32,33]. In turn, small sample sizes were also reported to affect the statistical power [30,31,32,34,35,36]. Convenience samples with self-engaged participants, hence motivated to participate in a survey, can also result in skewed findings [37]. Main reliance on snowballing procedures for participant identification [38], a selective experts’ engagement [39], and uneven distribution of participants by stakeholder type [40] or regions addressed [41] were also reported to affect the representativeness of the samples and the findings. Apart from cross-sectional studies, the single pilot trial included used a small sample of six occupational therapy students [34].

#### 3.1.2. Lack of Detailed, Up-to-Date, or Accessible Workforce Data

This category was addressed by seven papers. Lack of detailed workforce data was frequently identified. A secondary analysis of the occupational therapy workforce in South Africa was hampered by the lack of workforce data at multiple levels: e.g., employment by societal sectors (e.g., private, public, or non-governmental entities), practice sector (e.g., health, education, labor, or social development), practice areas (e.g., mental health), and identification of those not in practice due to unemployment, death, or retirement [42]. Other studies also identified a lack of detailed workforce data, especially for internationally trained occupational therapists [42,43,44]. Even when partly available, occupational therapy workforce data can be scattered across too many, unmerged data sources (e.g., from registration or licensure bodies, immigration agencies, population-level statistics, or sector-specific employment databases), creating difficulties in accessing, integrating, and readily comparing the information [42,43]. National workforce data is also sometimes non-comparable, collected by different methods by local structures [45] or by state licensure bodies that do not gather or report the same data on the same timelines [43,46]. Across nations, occupational therapy workforce data can be unavailable or inaccessible; although the WFOT regularly collects occupational therapy workforce data through the input of their member organizations, many are unable to provide or collect the data elements requested [47].

#### 3.1.3. Lack of Longitudinal Studies

This category was addressed by seven papers. A systematic review regarding the value of continuing professional development for recruitment or retention reported that the literature did not contain longitudinal studies and was limited to cross-sectional, exploratory, and descriptive survey designs [32]. Cross-sectional studies included reports that the nature of the study, even when coupled with advanced statistics, could not establish causal relationships or confirm the directionality of the associations [48,49,50,51,52]. Memory bias was another problem reported by cross-sectional studies, for example, for the measurement of attitudes before and after a program intervention [52] and conducting a retrospective evaluation of why providers left rural practice [53].

#### 3.1.4. Lack of Accounting for Local Market Needs and Dynamics

This category was addressed by six papers. A forecasting model assumed that the number of those entering or leaving occupational therapy is stable over time for all geographic areas, disregarding reasons for choosing or leaving the profession which may change over time; such changes may occur as the result of local market drivers and broader economic or policy changes [54]. Similarly, a US study regarding wage differentials among occupational therapists used aggregated data at the state level, not capturing local market dynamics [49]. Country-wide supply-need estimates were found to not account for patterns of supply and demand in local markets [55]. A prediction of workforce needs for scaling up rehabilitation services in Saudi Arabia recognized that using staff ratios from countries with highly developed services as benchmarks implied adaptation to the local context [56]. Finally, a multivariate regression model explained a mere 13% of all variance in the establishment of new occupational therapy positions across municipalities in Norway; factors such as the municipal economy, changing populations, or political roadmaps not considered in the analysis may also impact the variance [24].

#### 3.1.5. Participants from Single Contexts Impeding Generalizability or Sub-Group Analyses

This category was addressed by six papers. The studies had a focus or participants from a single context, such as a specific healthcare setting, an educational institution, a region, or sector (e.g., public versus private practices) [29,30,31,36,52,54]. This limitation affects the generalizability of the findings, as well as impedes sub-group analyses. For example, a study in the UK of strategies to recruit occupational therapists included managers only from public services, impeding generalization to the private sector or sub-group analyses [54]. As another example, a study of final-year occupational therapy students of a regional university in Australia focused on the willingness to practice in a rural area following participation in the educational program; such results are likely, program- or context-specific and not generalizable [52].

#### 3.1.6. Focus on Intentions and Self-Reports, Rather Than Behaviors or Occurrences

This category of limitations was addressed by four papers. A study on workforce retention acknowledged that reported intentions to leave were not the same as leaving practices [55]. A study on continuous professional development noted that perception or intentions to develop continuous professional development behaviors did not necessarily predict the adoption of such behaviors [56]. A survey study with similar limitations reported a focus on employment issues as self-reports, not as occurrences [24]. Finally, a study of occupational therapists’ perceptions of leadership styles reported that social desirability could play a role in the self-reported responses [29].

#### 3.1.7. Drawbacks of Comparisons of Multiple Professions

This category was addressed by three papers. Studies across multiple professions may lack interpretation, discussion of findings, or limitations that are specific to the occupational therapy workforce or its research [57,58], or may not clearly differentiate between therapists and therapy assistants [59].

#### 3.1.8. Challenges in Aggregating or Synthesizing Findings

This category was addressed by two papers. Findings of systematic reviews highlight difficulties with aggregating or comparing results of workforce research [32,60]. The literature was essentially descriptive with a lack of definition of variables impeding direct comparisons across studies [32]. Furthermore, the lack of working definitions for key terms under study, such as rurality, added to the complexity of knowledge synthesis [60]. In the conduct of the reviews, only one reviewer (a master’s student) was involved and worked under time constraints [32].

#### 3.1.9. Lack of Currency of Findings

This category of limitations was addressed by one paper. A study comparing continuing education requirements for licensure renewal across states acknowledged that findings may quickly become outdated because regulatory bodies continually modify and update requirements [39].

#### 3.1.10. Lack of Statistical Adjustment for the Testing of Multiple Associations

This category was addressed by one paper. One cross-sectional survey study acknowledged that the statistical analyses were not adjusted for the multiple associations tested (e.g., did not use Bonferroni correction), which leads to risks of false positives, i.e., type 1 error [51].

### 3.2. Research Recommendations

#### 3.2.1. Developing Context-Sensitive Comparisons: Across Professions, Geographies, and through Sub-Group Analyses

Recommendations for this category came from 10 papers. Workforce issues (e.g., supply, retention, compensation) might not be equal across countries and health professions, as well as within the same profession and country for rural and urban areas, across practice areas or settings, across regions of a country, and across strata of the occupational therapy workforce [36,41,49,61,62,63,64,65]. For instance, data on job satisfaction or career attraction for occupational therapists can be more relevant (e.g., reveal non-normative trends) when compared with that of other professions for the same context [36,65]. Further measuring, tracking, and regularly reporting on the extent of representation of diverse races/ethnicities is also recommended, and these results can encourage professional organizations, states, and individual institutions to make greater efforts to increase representation [42,64]. Studies using international standards or external benchmarks (e.g., on staff ratios) might develop research-based activities to adapt them to the local context and service delivery models or use proxy external comparators as benchmarks [66].

#### 3.2.2. Developing Longitudinal Studies That Include Interventions

Recommendations for this category came from seven papers. To overcome an over reliance on cross-sectional study designs and to help confirm the causality of the hypothesis raised by cross-sectional studies, more longitudinal research designs should be used [50,63,67]. Cohort study designs could monitor the workforce data for underserved areas [63] and also apply to trajectories over time, such as measuring how willingness for rural practice evolves with students through their occupational therapy education [52]. While retention issues were frequently addressed by cross-sectional studies, future work should focus on the design and evaluation of targeted programs or interventions to enhance retention determinants (e.g., job satisfaction, wellbeing, continuous development) and examine longitudinal effects on retention [31]. Studies using interrupted time series recommended follow-ups to understand changes in employment data patterns over longer periods of time and not only immediately after a critical event [68]. Pilot experiments of programs or interventions to increase recruitment or retention in underserved areas might also occur, and be followed by more solid experimental designs (e.g., with larger participant numbers, across multiple sites, with control groups, etc.) [34].

#### 3.2.3. Strengthening Routine Workforce Data Collection

Recommendations for this category came from six papers. If current and sufficiently detailed, workforce data collected by licensing and registration bodies or governmental agencies are preferred over incomplete information provided by professional associations with membership that is not mandatory [46,69]. Routine, comprehensive workforce data collection helps to monitor trends longitudinally [38,46,70], for a proactive rather than reactive response to workforce changes [46,71], and to provide a more solid determination of future shortages/surpluses [46,71,72].

#### 3.2.4. Studying the Impact on Ultimate Outcomes

Recommendations for this category came from five papers. In studies that analyze changes in employment trends, it is important to study whether any workforce reductions create efficiency improvements (e.g., reduce unneeded service utilization) or result in impeded population access to needed services or poor client outcomes [49,68,73]. New workforce tools (e.g., ePortfolio for continuous professional education) need to be evaluated not only in terms of their implementation but also in relation to the outcomes generated by the tools, for example, the impact or improvements in clinical practices secondary to continuous professional development behaviors [56]. Similarly, the evaluation of competency standards needs to address how the tools assist clinicians in their everyday clinical practice, not merely evaluate the development and implementation of the standards [39].

#### 3.2.5. Studying Evolving Demand in New Areas of Practice

Recommendations for this category came from five papers. As the demand for occupational therapists evolves, innovations in service delivery, payment reforms, and other health system-level factors might be considered for determining workforce requirements [46,71,74]. Policy changes, such as direct access to occupational therapy and demand from new settings of practice (e.g., primary care, community settings, health clubs, and senior centers), also might be considered in quantifying demand for the occupational therapy profession [72]. Research might assist in defining the role of occupational therapists in novel areas of practice. In addition, the acquisition of new roles (e.g., primary care practices) requires research to determine the necessary supporting regulations and financing arrangements [28].

#### 3.2.6. Promoting National Workforce Plans and Requirements

Recommendations for this category came from five papers. National occupational therapy workforce research needs to identify and seek solutions to overcome inequitable distributions of occupational therapists within a country, either per location (e.g., rural areas) or practice area (e.g., mental health). Development of clear staffing norms (e.g., minimum–maximum ranges) could provide a range of benchmarks that might assist with flexible implementation [42]. For retention in areas or geographies identified as underserved, national plans need to be developed and tested. These can go beyond solutions such as requiring community service to increase posts available in the public sector, reversing possible push factors (e.g., poor supervision or continuous education resources [42], lack of reimbursement for travel and telehealth [27]), and strengthening pull factors (e.g., positive fieldwork experiences, [63] recognising rural practice as a specialist field [27]). Similarly, national bodies could administer projects aiming to include occupational therapists in rural positions and build common competence-building projects, especially for therapists not employed in densely populated areas [24]. Task shifting to community health workers or other lower-level cadres in lower income countries can also extend access to services to the rural underserved population, provided these options are tested and that adequate supervision is in place [42]. Finally, workforce research should inform the work of regulatory bodies whose guidelines (e.g., for continuing education requirements) might be informed by an evidence base, not arbitrary or merely expert-based determination [75].

#### 3.2.7. Fostering International Coalitions for Workforce Research and Developments at Scale

Recommendations for this category came from two papers. Studies might be able to identify and replicate effective models of international cooperation and partnership on the training and education of occupational therapists for LMICs, including through inter-LMICs collaboration to increase training and education capacity for the scale-up of the occupational therapy workforce [69]. Cooperation between LMICs and developed countries also needs to be achieved to clarify the requirements to comprehensively inform those who may intend to practice internationally [76].

## 4. Discussion

The results of this scoping review synthesize the stated study limitations and recommended research reported by the occupational therapy workforce research, i.e., the 57 studies included in this scoping review. The limitations included a focus on cross-sectional studies with small and convenience samples; the use of participants from single settings or regions; a focus on self-reported data or information describing intentions rather than occurrences; the lack of statistical adjustment for testing multiple associations; and the lack of detailed, up-to-date, and accessible workforce data, to name a few. In turn, recommendations focused on developing longitudinal studies that include interventions (e.g., on recruitment or retention programs); strengthening routine workforce data collection; developing context-sensitive comparisons and sub-group analyses; studying the impact on ultimate outcomes; studying the evolving demand on new areas of practice; promoting national workforce plans and requirements; and fostering international coalitions for workforce research and developments at scale. This information can be used to guide the design of a WFOT-sponsored global strategy for the occupational therapy workforce research.

Not strangely, some of the reported recommendations directly address stated limitations. For instance, the recommendation on the further use of longitudinal designs, including on testing interventions or programs (e.g., recruitment and retention packages for underserved areas), resonates with an overreliance on cross-sectional designs and the lack of longitudinal studies, either observational or experimental. Similarly, the strengthening of routine workforce data collection can help address the lack of detailed, up-to-date, and accessible workforce data for immediate use in secondary research and longitudinal study designs. Having workforce research participants that are representative of the health workforce studied and of the population served by them is key for the generalizability of the findings and for a population-centered workforce research, policy, and planning [3,4,19].

While strengthening the collection of routine occupational therapy workforce data can be one potential leverage point for advancing the occupational therapy workforce research [19], doing so can be challenging without planned, concerted steps. Occupational therapists work across a multitude of societal sectors and employer types (e.g., health, social, educational, municipalities, non-governmental agencies), in varying practice areas including in new and emerging roles (e.g., primary care), with some countries relying on internationally-trained occupational therapists to meet internal demand [24,42,43,44,45,77,78]. When available, occupational therapy workforce data is scattered across too many, unmerged sector-based databases—with varying requirements, methods, and timings, even for data within the same sector [42,43,45,75]. Without a unified source of complete, current, and reliable workforce data, workforce research would be incomplete with strong limitations in interpretation and generalization of the findings. Workforce data from professional registration or licensing bodies can be a way to overcome the difficulties to track the occupational therapy workforce data [42,43,79]. However, using data from professional licensing bodies requires that these bodies (and broadly professional regulation) exist in the jurisdiction, have the means and framework to collect, update, and maintain the needed workforce data, and that the data requirements and collection methods are coordinated within a country and internationally [75].

National, coordinated plans, requirements, and activities can help harmonize workforce data collection across regional jurisdictions and establish benchmarks—albeit flexible—for the monitoring and promotion of equity in the distribution of occupational therapists. National plans can also promote (e.g., fund) the study and implementation of context-sensitive recruitment and retention programs targeting the underserved areas [80], as the funding rate for occupational therapy workforce research has been substandard [10]. Furthermore, national programs can provide the infrastructure and the scale for the continuous development and other activities on pull factors for occupational therapists working in underserved areas or smaller services; these can be a way to compensate for market inequalities and promote equity in population access for the occupational therapy workforce [24]. Overall, the national workforce research and development plans would need to target the occupational therapy workforce (i.e., with profession-specific components, data disaggregated by profession), even when part of broader rehabilitation, allied health, or health workforce plan and research. The literature reviewed here highlighted recommendations for cross-geography and cross-professional comparisons as one means to identify disparities but also identified disadvantages when data and implications are not disaggregated by profession type [57,58,59].

Finally, international coalitions are needed to underpin concerted occupational therapy workforce developments. While relevant across countries of varying income levels, international cooperation seems especially required for the scale-up of occupational therapy education and for workforce development in LMICs [69] and possibly in other countries where the occupational therapy profession and workforce seem under-developed (e.g., in Italy the proportion of occupational therapists per physical therapists is 2–98% when, for example, Israel nearly has a 50–50% distribution [81]). Cooperation can occur through LMICs within a region for scale and mutual learning, through the involvement of institutions and experts from higher-income countries, and both at the same time [82,83]. For example, the USAID funded the “SUDA project” (co-led by the World Confederation of Physical Therapists and Humanity and Inclusion) which implemented a capacity-building project in three French-speaking Sub-Saharan African countries; the project used international-level expertise to bolster professional associations, create resources sharing among stakeholders, and improved physical therapist educational standards in these countries [84,85]. Research and developments on strengthening associations and regulation for other professions also have been ramped up [86,87,88,89]. We are unaware of similar, externally funded workforce research and development projects for the occupational therapy field.

### Study Limitations

This scoping review synthesized the study limitations and recommendations as reported by the reviewed literature; therefore, they do not reflect a methodological appraisal of the included studies. Hence, the limitations and recommendations reported here are not necessarily exhaustive or examined beyond the peer-reviewed process that led to the studies’ publication. Similarly, although we provide the number of papers addressing each type of study limitation (e.g., on the lack of statistical adjustment for the multiple associations tested), these numbers reflect the frequency of reporting of these limitations, not necessarily the frequency of their occurrence. Occurrence is possibly under-reported; hence, the results might be understood especially by the type and range of study limitations and recommendations and not necessarily by the frequency of their report. To overcome these limitations, we plan to convene an interdisciplinary, interprofessional panel of health workforce experts, representatives of different geographies, to collectively analyze the strengths, weaknesses, and proposed directions for the occupational therapy research worldwide based on these findings, a quantitative map (e.g., study methods, geographies, funding rates) of the literature [10], and qualitative analyses of the findings type [11], which referred to the three types of results from this scoping review project. Methodologically, we applied a temporal cut-off to restrict the study to the literature published in the last 25 years; therefore, the results of the review do not include data for older papers. Furthermore, the second data-extractor role was performed by three different researchers, a practice which can add bias. Finally, the database searches were only run in June 2021; hence, references published beyond this point in time were included only if identified though the iterative snowballing process and key-informant recommendations.

## 5. Conclusions

These results of this large-spectrum scoping review specifically address the reported study limitations and recommended research of the occupational therapy workforce research. Results suggest the need to strengthen routine workforce data collection and move beyond cross-sectional studies toward embracing longitudinal and experimental studies (e.g., that address occurrences beyond intentions, including ultimate outcomes, and test recruitment and retention packages for underserved areas). These may be complemented by national plans and international coalitions, with frameworks, coordination, and a scale for studying and developing a sizeable, equitably distributed, and competent occupational therapy workforce that meets local and global population needs. Using the limitations and recommendations here identified, the WFOT is planning to develop a stakeholders consultation process toward identifying and refining global strategic directions for advancing the occupational therapy workforce research and development worldwide. The wide endorsement and implementation of these strategies might strengthen the profession’s ability to equitably meet the population health, rehabilitation, and occupational needs.

## Figures and Tables

**Figure 1 ijerph-19-07327-f001:**
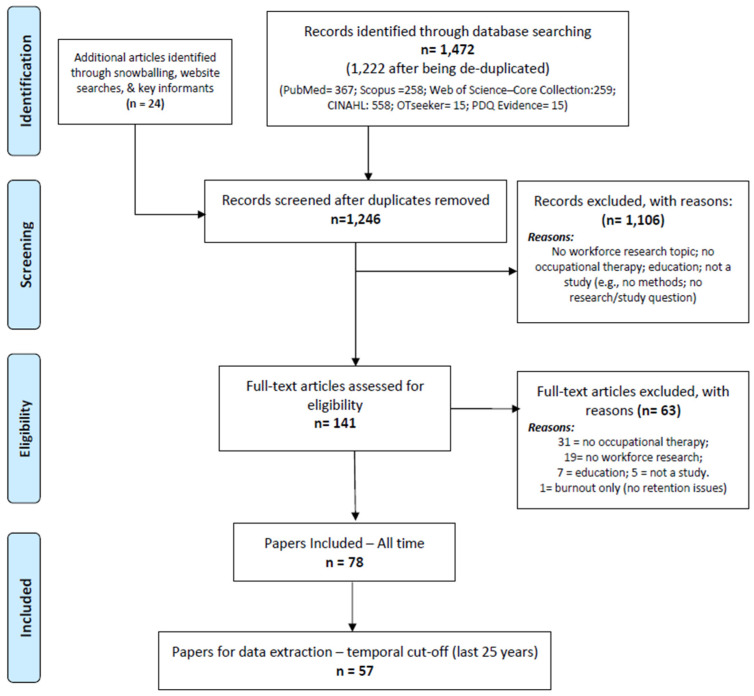
PRISMA flowchart of the scoping review.

**Table 1 ijerph-19-07327-t001:** Inclusion categories for the major topics of workforce research included, synthesized from the review protocol.

Inclusion Category	Category Type
1	Workforce supply (e.g., supply of practicing therapists or mapping their profile)
2	Workforce production (e.g., graduates supply or entry-level requirements)
3	Workforce needs, demands, or supply-need/demand shortages; forecasts
4	Employment trends (e.g., (un)employment patterns, unfilled vacancies)
5	Workforce distribution (e.g., per geographies, practice area, public vs. private sectors)
6	Geographical mobility (e.g., (e/im) migration; internationally trained workers)
7	Attractiveness and retention (e.g., salaries, incentives, job satisfaction, intention to leave the profession, recruitment determinants)
8	Staff management and performance (e.g., human resources management, workload management, recruitment practices from managers, staffing and scheduling, burnout associated to performance or productivity)
9	Regulation and licensing (e.g., continuing education requirements, task shifting, evaluating the impact of licensing or regulatory changes)
10	Systems-based or systematic analysis of workforce policies

## Data Availability

This study used published data.

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
