# Peer review of "Limitations and Recommendations for Advancing the Occupational Therapy Workforce Research Worldwide: Scoping Review and Content Analysis of the Literature"

_ijerph, 2022, doi:10.3390/ijerph19127327_

Round 1
Reviewer 1 Report
Abstract:
In line 22, the authors need to indicate which investigations they are referring to in the second sentence of the abstract.
Introduction
In line 41, the authors need to clarify whether it is supposed to be strength or strengthening so that the sentence will read well.
In line 69 -70, we aim to synthesize the limitations and future recommendations that were included in the papers reported about the advancement of the occupational therapy workforce research.
Materials and Methods:
In line 75, the authors wrote about the exploratory research questions. However, they did not indicate the research questions.
The authors highlighted that they have used conventional content analysis, which is inductive in nature.
Results
The authors used categories that emerged in their analysis to present the results of the scoping review. Each category was also accompanied by the frequencies of the studies that reported on the specific results.
Discussion:
The discussion section is structured based on the recommendations that were found to be relevant to address the limitations of the studies included in the scoping review.

Author Response
We submit here the response to all 3 reviewers.

Reviewer 2 Report
The present manuscript aims to find the limitations that impede Occupational Therapy to reach a higher development. With this objective, the authors conducted a scoping review looking for gaps of knowledge to build recommendations. The study is well written, the results and discussion are constructive but some details should be addressed specially in methods section before consider its publication:
Line 1: Remove “type of paper”
Line 49: remove coma before “and”
Line 54: remove coma before “and”
Line 85: why wasn’t the review updated since June 2021? There is near a year since then.
Line 85: supplementary materials were not supplied to reviewer. Almost I am not able to find them in the platform.
Line 120: the application of a saturation cut of is a bias that should be included in limitations section.
Line 126: how was the information extracted? Xls table? Sheet teport? What did the authors in case of disparities among researchers at extracting data? Why was not the same researcher always in charge of verify data extracted by TJ? The criteria could have change among researchers.
Figure 1: more quality is needed, the words seem pixeled.
Author Response

(The authors gave the same response as above.)

Reviewer 3 Report
This scoping review complements a multifaceted review led by the World Federation for Occupational Therapy (WFOT), with a particular interest in the discipline. The first publication focused on the quantitative trends of the subject and the second showed a synthesis of the results obtained. The special interest of this third work is that it makes an in-depth analysis of the limitations in research related to the occupational therapy workforce, and consequently the specification of recommendations. This information may have an impact on future WFOT strategies and research on the occupational therapy workforce worldwide.
The scoping review follows the Arks & O'Malley framework and the Joanna Briggs Institute Guidelines, presenting essential information throughout the process.
Reviewing that all the limitations identified are all present in the abstract, I think it would be interesting to include some reference to 3.1.8, perhaps including the concept of descriptive studies at some point in the narrative, to point out the impossibility of aggregating and comparing results. Although only two papers talk about it, in the abstract the lack of statistical adjustment is given prominence, which is only commented on in one paper.
When you state in lines 414 and 415 "We are not aware of similar workforce research and development projects for the field of occupational therapy," I suggest doing a search for more local projects that may have a similar focus to the SUDA project, but in occupational therapy. Considering the relationship between practice, education, and research in occupational therapy, I suggest linking these different areas in the discussion and mentioning some projects that can be inspiring for possible projects and can have an impact on clinical practice. The good practices of the European Network of Occupational Therapy in Higher Education (ENOTHE) in the educational field can be an example:
Van Bruggen, H. (2011). Eastern European countries in transition: capacity building for social reform. Occupational Therapies Without Borders, 2, 295-303.
Van Bruggen, H. (2014). Turning Challenges into Opportunities: How Occupational Therapy is Contributing to Social, Health, and Educational Reform. Bulletin of the World Federation of Occupational Therapists, 70(1), 41-46.
Author Response

(The authors gave the same response as above.)

Round 2
Reviewer 2 Report
Thank you.